# Ultrastructural Alterations of the Human Pathogen *Giardia intestinalis* after Drug Treatment

**DOI:** 10.3390/pathogens12060810

**Published:** 2023-06-07

**Authors:** Marlene Benchimol, Ana Paula Gadelha, Wanderley de Souza

**Affiliations:** 1BIOTRANS-CAXIAS, Universidade do Grande Rio. UNIGRANRIO, Rio de Janeiro 96200-000, Brazil; 2Instituto de Biofísica Carlos Chagas Filho, Universidade Federal do Rio de Janeiro, Rio de Janeiro 21941-901, Brazil; wsouza@biof.ufrj.br; 3Instituto Nacional de Ciência e Tecnologia em Biologia Estrutural e Bioimagens e Centro Nacional de Biologia Estrutural e Bioimagens, Universidade Federal do Rio de Janeiro, Rio de Janeiro 21941-901, Brazil; 4Diretoria de Metrologia Científica, Instituto Nacional de Metrologia, Qualidade e Tecnologia (INMETRO), Rio de Janeiro 25259-020, Brazil; anagadelha@gmail.com

**Keywords:** intestinal parasite, diarrhea, *Giardia* treatment, children’s disease

## Abstract

This review presents the main cell characteristics altered after in vitro incubation of the parasite with commercial drugs used to treat the disease caused by *Giardia intestinalis*. This important intestinal parasite primarily causes diarrhea in children. Metronidazole and albendazole are the primary compounds used in therapy against *Giardia intestinalis*. However, they provoke significant side effects, and some strains have developed resistance to metronidazole. Benzimidazole carbamates, such as albendazole and mebendazole, have shown the best activity against *Giardia*. Despite their in vitro efficacy, clinical treatment with benzimidazoles has yielded conflicting results, demonstrating lower cure rates. Recently, nitazoxanide has been suggested as an alternative to these drugs. Therefore, to enhance the quality of chemotherapy against this parasite, it is important to invest in developing other compounds that can interfere with key steps of metabolic pathways or cell structures and organelles. For example, *Giardia* exhibits a unique cell structure called the ventral disc, which is crucial for host adhesion and pathogenicity. Thus, drugs that can disrupt the adhesion process hold promise for future therapy against *Giardia*. Additionally, this review discusses new drugs and strategies that can be employed, as well as suggestions for developing novel drugs to control the infection caused by this parasite.

## 1. Introduction

### Giardia Intestinalis and Its Biological Cycle

Giardia is an extracellular parasite that colonizes and proliferates in the host’s small intestine, leading to the diarrheal disease known as *Giardiasis* [1]. While it can be found in various vertebrates, including humans and domestic animals, a recent study reported the presence of Giardia-like trophozoites in the gut of the invertebrate *Heterotermes tenuis* [2].

Based on biochemical and molecular studies and electron microscopy descriptions, six species have been identified, including *G. intestinalis* (also known as *G. duodenalis* and *G. lamblia*), which infects mammalian hosts. Although these organisms exhibit morphological similarities, molecular biology approaches have unveiled genetic diversity within the *G. intestinalis* species [3]. It has been demonstrated that *G. intestinalis* isolated from humans can be classified into two main genetic groups: A and B assemblages [4]. These genetic variations have been linked to differences in metabolism, in vitro growth rates, and infectivity. Furthermore, additional assemblages have been identified, such as the C and D assemblages found in dogs; and the E, F, and G assemblages found in cattle, cats, and rats, respectively. The H assemblage is specific to marine mammals [5]. Studies on genetic diversity play a crucial role in understanding the potential zoonotic transmission of the parasite, as some animals may be involved in outbreaks of human *Giardiasis*.

Infection occurs when an individual ingests cysts, which can be present in contaminated water or food. Other modes of transmission include person-to-person contact through unwashed hands and exposure to contaminated sand or toys in playgrounds. The stomach’s acidic environment and the action of intestinal proteases trigger the excystation process, leading to the emergence of two trophozoites from a single cyst. These trophozoites adhere to the intestinal epithelial cells in the duodenum and the initial portion of the jejunum (Figure 1). Through binary fission, they multiply and establish parasitic colonization in the gut. Encystation occurs when trophozoites pass through the intestinal lumen and reach the large intestine. During the trophozoite-to-cyst differentiation, various changes occur, including synthesizing a cyst wall. Finally, mature cysts are excreted in the feces, capable of infecting new hosts and initiating the cycle anew.

## 2. Morphology of the Two Stages in the Life Cycle of *Giardia intestinalis* as Observed Using Light and Electron Microscopy

*Giardia* presents two developmental stages: the cyst and the trophozoite.

### 2.1. Trophozoite

The trophozoite of *G. intestinalis* presents four pairs of flagella and two nuclei (Figure 2 and Figure 3). The trophozoite measures 9 to 21 μm in length and 5 to 15 μm in width, with a 2 to 4 μm thickness. Its anterior region is larger, while the caudal region is narrow, giving the cell a pear-shaped appearance. The eight flagella originate from the basal bodies and emerge in pairs at various positions. A notable feature of this parasite is the presence of a ventral disc composed of microtubules and microribbons. Additionally, the parasite exhibits a cytoplasmic lip known as the flange.

Moreover, the parasite possesses other structures, such as the median body and funis, which are composed of microtubules, making them valuable subjects for research. The primary organelles found in trophozoites include peripheral vesicles, mitosomes, and endoplasmic reticulum. Additionally, glycogen granules and ribosomes are present. Figure 3 provides an overview of the general appearance of *Giardia intestinalis*.

#### 2.1.1. Nuclei

*G. intestinalis* trophozoites possess two spherical or oval nuclei positioned symmetrically above the ventral disc (Figure 3a). Each nucleus contains five chromosomes and a comparable amount of DNA.

#### 2.1.2. Flagella

In *Giardia*, the eight flagella consist of anterior, posterior, ventral, and caudal flagella (Figure 2 and Figure 3b) [7]. These flagella are believed to contribute to the parasite’s motility in multiple directions and help in the adherence of trophozoites to the host’s intestinal epithelial cells [8,9].

#### 2.1.3. Ventral Disc

Microtubules play a crucial role in forming the ventral disc, facilitating the parasite’s adhesion to host cells and to glass or plastic substrates (Figure 1, Figure 2 and Figure 3) [10,11,12]. Occupying approximately two-thirds of the trophozoite’s anterior region (Figure 3b), the disc consists of coiled microtubules that run parallel to the ventral plasma membrane. Extending from these microtubules are microribbons, fibrous structures composed of three parallel sheets [13,14]. Adjacent microribbons are connected by electron-dense structures referred to as cross-bridges, and they contain proteins known as giardins. Two types of giardins have been identified: alpha–beta- and gamma-giardins. Notably, the gamma-giardin lacks homology with other proteins [7]. Lastly, the disc contains a central region known as the “bare area”, which lacks the microtubule–microribbon complexes.

#### 2.1.4. Median Body

The median body, a structure formed by microtubules in the posterior half of the cell, is present in approximately 80% of the cells and serves as a valuable characteristic for taxonomic studies. Rather than being a singular or dual structure, the median bodies vary in number, shape, and position. The microtubules within the median body are connected to the plasma membrane, adhesive disc, or caudal flagella. Although not entirely free within the cells, they can extend and protrude on the cell surface. Furthermore, the microtubules of the median body exhibit reactivity with several anti-tubulin and anti-beta-giardin antibodies [15].

#### 2.1.5. Peripheral Vesicles and the Endocytic Pathway

Peripheral vesicles are organelles positioned just beneath the plasma membrane, with dimensions of approximately 150–200 nm (Figure 3a). These vesicles correspond to the endosomal–lysosomal systems found in other eukaryotic organisms [16]. Apart from their involvement in endocytic processes, these vesicles also contain multivesicular bodies, indicating a potential functional role in exocytic activity [17].

#### 2.1.6. Endoplasmic Reticulum and Golgi Complex

*G. intestinalis* possesses a functional endoplasmic reticulum (ER) (Figure 3a) composed of numerous cisternae. The ER serves as a site for protein synthesis, endocytosis, and material degradation [18]. A tubulovesicular network has been observed, indicating communication between the clathrin-coated peripheral vesicles and the endoplasmic reticulum [18]. Moreover, the ER undergoes expansion during encystation [19]. However, no canonical Golgi apparatus has been identified in *G. intestinalis*.

#### 2.1.7. Mitosomes

*G. intestinalis* lacks mitochondria but possesses a double-membrane-bound organelle called the mitosome, which measures 150–200 nm [20]. The mitosome does not perform various mitochondrial functions, including ATP synthesis, the citric acid cycle, lipid metabolism, heme biosynthesis, and the urea and amino acid cycles. Additionally, DNA has not been detected within the mitosome [20]. However, it does contain protein machinery components responsible for iron–sulfur (Fe–S) cluster assembly, such as the IscS and IscU proteins [21], as well as other proteins such as chaperonin Cpn60 and mitosomal HSP70 [22]. Furthermore, the mitosomes possess protein import systems known as TIM and TOM, which are characteristic of mitochondria.

### 2.2. Cyst

*Giardia* exhibits a cystic form (Figure 4) that resists harsh environmental conditions, allowing it to remain viable for several months. These cysts are oval-shaped, measuring 8 to 12 µm in length and 7 to 10 µm in width. They are enclosed by a cyst wall that is 0.3–0.5 µm thick, consisting of an outer filamentous layer and an innermost layer composed of two membranes [23]. Furthermore, the flagella become internalized within the cyst, and the adhesive disc becomes fragmented. Multiple nuclei, ranging from two to four, are also present within the cyst.

Trophozoites undergo encystation in response to adverse conditions, such as nutrient deprivation or migration to the lower portions of the ileum and large intestine [24]. This process can also be induced in vitro. During encystation, trophozoites detach from the intestinal epithelium, internalize their flagella, and gradually adopt an oval shape [25]. Notably, encystation secretory vesicles (ESVs) emerge within the cytosol. These vesicles are characterized by their large size, morphological irregularity, and the presence of cyst wall proteins (CWPs) [26].

The excystation process takes place when the cyst encounters favorable environmental conditions. *Giardia* proteases break down the cyst wall, starting at the pole of the cell. Through this opening, the flagella emerge, and their beating appears to aid in the emergence of a tetranucleated oval cell. This cell gradually elongates, flattens, and undergoes cytokinesis, forming two trophozoites [27].

## 3. Compounds Used in the *Giardiasis* Treatment and Their Effects on the Parasite (Table 1)

### 3.1. Metronidazole

The treatment of *Giardiasis* involves administering compounds from the nitroimidazole class, which are effective against various bacterial and protozoan infections. The first nitroimidazole compound to be discovered was azomycin, a 2-nitroimidazole isolated from actinobacteria. Further research led to the development of a synthetic 5-nitroimidazole called metronidazole, commonly used to treat trichomoniasis [28]. Subsequently, Darbon et al. [29] reported that this compound could also be used to treat *Giardia* infections. Other nitroimidazoles, such as secnidazole and tinidazole, have demonstrated clinical efficacy [30,31]. Among the nitroimidazoles, metronidazole is the most extensively studied in terms of its mechanism of action against *G. intestinalis*, and it is the most commonly prescribed medication for *Giardiasis*.

Upon entering the parasite, metronidazole undergoes activation through the donation of electrons to its nitro group by enzymes such as pyruvate:ferredoxin oxidoreductase (PFOR), thioredoxin reductase, or nitroreductase [32]. This drug reduction process forms nitro radical anions and subsequently generates hydroxylamine, amine, and superoxide. In vitro studies have demonstrated that the drug primarily affects the replication phase of the *G. intestinalis* cell cycle and induces DNA fragmentation, which is considered a secondary effect resulting from damage to other biomolecules [33]. Electron microscopy analysis revealed (Figure 5) no significant changes in the nuclei’s ultrastructure. However, there was an observed increase in the size of peripheral vesicles and the presence of myelin-like figures in the cytoplasm [34]. Furthermore, proteomic approaches have shown that metronidazole can bind to various proteins, including beta-giardin and elongation factor 1 [35]. *G. intestinalis* cysts exhibit reduced susceptibility to metronidazole, but residual bodies can still be found in the nuclei [36]. In vitro studies have induced drug resistance, which initially appeared to be associated with decreased PFOR levels, but subsequent research has revealed the involvement of other mechanisms, such as dysfunctional nitroreductases [37]. Despite its extensive global use for treating various diseases, the “US Food and Drug Administration” has not approved its indication for treating *Giardiasis* [38].

### 3.2. Albendazole

Other compounds, such as those belonging to the benzimidazole class, have been utilized to treat *Giardiasis*. These drugs have been extensively employed as anthelmintic agents in veterinary and human medicine since the 1960s, and they have also been used as antifungals in agriculture. In the 1980s, their inhibitory effects on the growth of *Trichomonas vaginalis* were discovered [39]. Subsequently, similar effects were observed against *G. intestinalis* [40]. Despite demonstrating efficacy in vitro, clinical treatment with benzimidazoles has yielded conflicting results, with lower cure rates than those achieved with metronidazole. As a result, these compounds are considered a secondary option for treatment.

Analysis of the beta-tubulin sequence (residues 108 to 259) revealed that *G. intestinalis* and *T. vaginalis* possessed two sites (Glu-198 and Phe-200) that rendered them susceptible to benzimidazoles. In contrast, *Entamoeba histolytica* and *Leishmania major* exhibited resistance to benzimidazoles, as they had different amino acids in these regions of the molecule [41]. Benzimidazoles are known to bind to the parasite’s beta-tubulin, creating a “cap” at the plus end of microtubules [42].

Albendazole, a benzimidazole carbamate, exhibits activity against *G. intestinalis*, and trophozoites can metabolize this drug into its toxic intermediates: albendazole sulfone and albendazole sulfoxide. The conversion process involves a recently described flavohemoglobin present in the parasite [43]. Despite the fact that several microtubular structures constitute the parasite’s cytoskeleton, electron microscopy studies have shown that the drug specifically affects the ventral disc (Figure 6) [44]. Consequently, the parasite’s adhesion is compromised due to disc fragmentation. Furthermore, albendazole can potentially interfere with mitotic spindle development during cell division, particularly affecting the G2 phase of the *G. intestinalis* cell cycle [33]. In vitro studies have demonstrated the ability to induce albendazole resistance, which correlates with beta-tubulin mutation, modulation of enzymes involved in glycolysis and arginine metabolism, and decreased *Giardia* flavohemoglobin mRNA expression [43,45,46].

### 3.3. Other Compounds

Metronidazole and albendazole remain the primary compounds employed in the therapy used against *Giardia intestinalis*. However, additional compounds such as nitazoxanide, furazolidone, and paromomycin are also utilized.

Nitazoxanide, a nitrothiazolide compound, exhibits activity against bacteria, protozoa, and helminths. It was first approved for the Treatment of *Giardia* infections in 2004. The drug’s mechanism of action is believed to involve inhibiting enzyme activities in the parasite, such as PFOR and nitroreductases [47]. Another hypothesis suggests that the drug undergoes reduction of the nitro group, similar to metronidazole, although this is still uncertain. Electron microscopy analysis (Figure 7) demonstrates that the parasite’s ventral disc and plasma membrane are affected, indicating potential impairment of adhesion following treatment [48]. Interestingly, nitazoxanide also damages the cyst wall by damaging the *G. intestinalis* cyst [36]. Resistance to nitazoxanide, as observed with metronidazole, may be associated with alterations in nitroreductase activity [47].

Furazolidone (Figure 8) is another drug used to treat *Giardiasis*. It has been effective against bacteria and utilized for *Giardiasis* treatment since 1950. Furazolidone is approved for use in the United States and is considered an important drug in several countries. It is administered in suspension form and is widely employed in pediatric cases [49].

The action mechanism of furazolidone is similar to that of metronidazole, as it also targets the parasite’s electron transport pathway. However, furazolidone is reduced by NADH oxidase instead of PFOR [50]. Due to its chemical structure, this drug contains a nitro radical, producing toxic intermediates. These secondary metabolites of furazolidone can cause damage to various cell components, including DNA [50]. Moreover, furazolidone alters the ultrastructure of the protozoan (Figure 8), depleting cytoplasmic contents and reducing the parasite’s adherence and proliferation [34].

In vitro studies have demonstrated that parasite resistance to furazolidone can be induced and is correlated with increased levels of thiol-dependent enzymes [51].

Paromomycin, an aminoglycoside, demonstrates in vitro efficacy against *G. intestinalis*. It binds to ribosomal subunits, thereby inhibiting protein synthesis [52]. Despite its favorable activity, clinical treatment with this drug has yielded lower cure rates than metronidazole, albendazole, and other compounds [53]. Consequently, the use of paromomycin is restricted to pregnant women due to the unknown effects of nitroimidazoles during the first trimester of pregnancy, while albendazole is considered teratogenic.

Quinacrine, an acridine compound with a broad spectrum of activity against various pathogens, was historically used to treat *Giardiasis*. However, its utilization has been limited due to side effects such as vomiting and psychosis. Currently, quinacrine is reserved for cases that do not respond to metronidazole treatment in certain countries [54]. In addition, this drug’s precise mechanism of action remains unclear, although it is believed to potentially inhibit parasite DNA synthesis [49].

**Table 1 pathogens-12-00810-t001:** Drugs affecting *G. intestinalis* used in therapy.

Drug	Mechanism	In Vitro Effects on the Parasite	Reference
Metronidazole 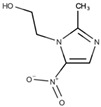	Reduced nitro group; forms toxic intermediate	DNA fragmentation Autophagic-like vacuoles Cell cycle arrest in the S phase	[33,34]
Albendazole 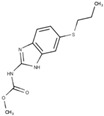	Bind to the beta-tubulin forming a “cap” at the plus end of microtubules	Ventral disc disrupted Cell cycle arrest in the S phase	[44]
Nitazoxanide 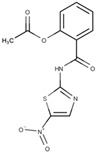	Inhibition of enzyme activity (PFOR, nitroreductase)	Plasma membrane disruption Ventral disc fragmented	[48]
Furazolidone 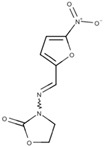	Reduced furazolidone; forms a toxic intermediate	Depleted cytoplasm Autophagic-like vacuoles	[34]
Paromomycin 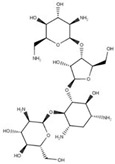	Inhibit protein synthesis	No described	[52]
Quinacrine 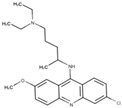	Inhibit DNA synthesis	No described	[49]

## 4. Potential Drug Target

Despite the compounds mentioned above (item 3) being the most commonly used in therapy, treatment failures and high recurrence rates suggest the presence of parasite resistance to these drugs [55]. Furthermore, the activity of these compounds is still being investigated due to reports of their cytotoxicity, mutagenic potential, and carcinogenic properties [56]. Pharmaceutical industries have not prioritized the development of new antiparasitic agents, resulting in the continued administration of these compounds for over 50 years. As a result, the search for alternative therapies to treat *Giardia* infections becomes increasingly important. Numerous studies have explored novel compounds, primarily through drug repositioning, leading to significant findings that may serve as a foundation for the future synthesis of new anti-*Giardia* medications.

### 4.1. Giardia Cytoskeleton—Drugs Affecting Microtubules: Nocodazole, Colchicine, Taxol, Vinblastine, and Sulfonamide Oryzalin

A notable characteristic of the *G. intestinalis* cytoskeleton is the presence of microtubules, which organize into distinct structures such as the ventral disc, median body, and funis. The parasite is also equipped with four pairs of flagella featuring a 9 + 2 arrangement of microtubules and associated extra-axonemal structures [8,57]. Given the intricate nature of this microtubular cytoskeleton, compounds that disrupt tubulin dynamics serve as crucial therapeutic strategies, as they directly impact essential biological processes, including motility, adhesion, and cell division.

Over the years, numerous compounds targeting microtubule polymerization/depolymerization have been evaluated against the parasite. Alongside albendazole, other benzimidazole derivatives have demonstrated notable efficacy. Among them, nocodazole, a methyl [5-(2-thienyl carbonyl)-1H-benzimidazol-2-yl]carbamate, is known to bind to beta-tubulin, resulting in the depolymerization of microtubules [58]. Exposure of *G. intestinalis* to this compound leads to the inhibition of trophozoite proliferation and accumulation of cells in the G2 phase of the cell cycle [58]. Ultrastructural analyses reveal the impact on parasite adhesion, with a disrupted ventral disc (Figure 9). The size of the median body decreases, and electron-dense deposits of tubulin are observed within the cytoplasm [58]. The structure of the flagella remains unaltered, although there is an increase in flagellar length [59].

The efficacy of another member of the benzimidazole class, the non-carbamate benzimidazole thiabendazole, has also been assessed against *G. intestinalis*. This compound inhibits parasite proliferation and adhesion, albeit less effectively than albendazole and mebendazole. Therefore, it is considered a drug with moderate activity [60].

The effects of colchicine, an agent targeting microtubules, were also investigated. Colchicine binds to soluble tubulin, forming a tubulin–colchicine complex that can hinder microtubule elongation by binding to their plus ends. When trophozoites were exposed to colchicine, similar changes to those observed with nocodazole were observed (Figure 9 and Figure 10), albeit at longer time intervals and higher concentrations (>50 μM) [58]. However, colchicine did not affect the cell cycle of *G. intestinalis*.

Taxol, vinblastine, and the sulfonamide oryzalin, a nitroaniline herbicide that binds to tubulin, have been examined for their effects on parasites due to their impact on microtubules. These compounds reduce parasite proliferation, although high drug concentrations are necessary to induce trophozoite death. Moreover, taxol and oryzalin lead to an increase in median body size and cause flagella shortening [59,61] (Figure 11). Treatment with oryzalin prevents parasites from completing cell division [61].

Mebendazole, a benzimidazole carbamate compound, exhibits 30–50 times higher activity than metronidazole against *Giardia*. Furthermore, it has been shown to cause damage to the ventral disc of trophozoites [40]. While the previously mentioned compounds demonstrate some in vitro effectiveness, albendazole and mebendazole are less toxic and display superior efficacy in both in vitro and in vivo analyses. Therefore, these drugs are considered more suitable for the Treatment of Giardiasis.

### 4.2. Giardia Cytoskeleton—Drugs Affecting Actin: Cytochalasin D, Latrunculin A, Jasplakinolide

The organization and functional role of actin in *Giardia* has been controversial for many years due to its highly divergent nature. However, recent studies indicate that this protein can form filaments and is associated with cell division and endocytosis [62]. Consequently, compounds that interfere with actin dynamics hold promise as potential anti-*Giardia* drugs. Cytochalasin D, latrunculin A, and jasplakinolide have been tested, yielding interesting results. These compounds generally inhibit in vitro parasite proliferation, with cytochalasin D binding to actin and exerting a stronger inhibitory effect on microfilament polymerization and elongation, typically at concentrations around 10 μM [62,63]. Electron microscopy studies have investigated its mechanism of action on the parasite, revealing intriguing effects such as the formation of aberrantly shaped, multiflagellated cells (Figure 12), indicating an inability of the trophozoite to complete the cytokinesis process [63].

### 4.3. Giardia Organelles Affected by Drugs (Table 2)

Certain organelles, such as peripheral vesicles and mitosomes, exist in *G. intestinalis* but are not observed in higher eukaryotic cells, despite potentially performing functions already present in other organisms. Currently, no drugs have been specifically designed to target these organelles’ structure or functional role. However, electron microscopy studies reveal that metronidazole and pyrantel pamoate affect these structures. After incubation with these drugs, the size of peripheral vesicles increases, although the exact mechanism of action remains unclear [34,64]. Furthermore, compounds such as cytochalasin D and the anti-microtubule agent vinorelbine interfere with the endocytosis process, which is one of the functions associated with these vesicles [65]. Nevertheless, drugs specifically targeting the structures or functions of this organelle have not yet been reported, thus presenting new avenues for further investigation.

### 4.4. Drugs Inducing Cell Death

Although *Giardia* lacks a typical mitochondrion, characteristics of cell death have been observed in the parasite following drug treatment. For instance, beta-lapachone, an ortho-naphthoquinone used against various microorganisms, induces abnormal shaping of the parasite, with plasma membrane blebs (Figure 13), cytoplasmic vacuolization, and chromatin condensation all indicative of an apoptotic process [66]. Other typical features, such as phosphatidylserine exposure on the outer cell membrane, have been observed after treatment with linear lactone, kaempferol, lactoferrin peptides, and 1-hydroxy-2-hydroxymethylanthraquinone. DNA fragmentation, a common characteristic of apoptotic cells, has also been documented in parasites treated with metronidazole and KH-TFMDI, a sirtuin inhibitor [34,67]. Published studies suggest that the activation of apoptosis-like cell death in *G. intestinalis* differs from the classical mechanism and occurs through a caspase-independent pathway [67]. Since the entire apoptotic machinery is absent in protozoa, alternative forms of programmed cell death have been proposed [68].

### 4.5. Drugs Affecting Carbohydrates and Lipids Metabolism

Due to the absence of typical mitochondria, *Giardia* relies on substrate-level phosphorylation for its metabolism [69]. Consequently, glycolysis plays a crucial role in ATP production within this parasite, and the conversion of glucose to pyruvate through the glycolytic pathway occurs outside of a specific organelle [69]. The enzymes involved in this process can possess distinct characteristics from those found in mammalian cells, making them potential targets for drug intervention. Triosephosphate isomerase is one example responsible for converting (D)-glyceraldehyde-3-phosphate to dihydroxyacetone phosphate. Recent studies have indicated that this enzyme in the parasite is inhibited by omeprazole, an H+/K+ATPase inhibitor, and disulfiram, a cysteine modifier agent [70,71]. The mechanism of action of omeprazole has been investigated through light and electron microscopy (Figure 14). The results revealed the presence of lamellar structures, vesicles, and increased glycogen deposition in treated parasites, suggesting a potential disruption of energy generation [71].

In *G. intestinalis* trophozoites, the synthesis of membrane lipids involves the uptake of fatty acids from the medium, which are then incorporated into various phospholipids. Fatty acid remodeling occurs through the activities of elongases and desaturases [72]. The sterol biosynthesis pathway in *Giardia* is limited and includes the production of isoprenoids and dolichol. Cholesterol is the primary sterol present in *Giardia* membranes.

Several compounds targeting metabolic pathways involved in sterol synthesis have been tested against *Giardia* in vitro. For instance, azasterol and epiminolanosterol, which inhibit Δ24(25)sterol methyl transferase (24-SMT), have demonstrated activity against trophozoites, despite the parasite not producing 24-alkyl sterols or ergosterol. Treatment with these drugs showed parasite proliferation and enlargement of peripheral vesicles and encystation vesicles (Figure 15) [73]. An evaluation was also conducted on ibandronate and risedronate, compounds that affect farnesyl diphosphate synthase, an enzyme involved in the biosynthesis of isoprenoids, dolichols, and cholesterol. Although this enzyme is present in *G. intestinalis*, these compounds exhibited low activity and only showed effects at concentrations exceeding 200 μM [74]. Cytoplasmic vacuolization was observed in cells treated with ibandronate, as shown in Figure 16.

### 4.6. G. intestinalis Kinases as Target Drug

The *Giardia* kinome consists of 278 proteins, with 80 considered the core kinome. Out of these kinases, 61 can be classified into families that have already been described in higher eukaryotes. The remaining 19 kinases either belong to specific families of *Giardia* or show no homology to any other protein [75]. Additionally, the parasite possesses NEK kinases, constituting 198 out of the 278 identified protein kinases. Most NEK kinases appear to serve a structural function and are localized in cytoskeletal elements. Some are active during phases such as mitosis and excystation [75].

Given the highly divergent sequences and distinct families within the *Giardia* kinome, it is expected to emerge as a specific target for drug development. Some kinase inhibitors have already undergone testing against parasites, yielding interesting results. For instance, aurora kinase inhibitors reduced parasite proliferation and compromised cytokinesis [76]. Similarly, screening-bumped kinase inhibitors, originally designed to inhibit calcium-dependent protein kinases of apicomplexan pathogens, revealed five compounds that interfered with *G. intestinalis* growth. Among them, one compound (BKI1213) specifically affected cytokinesis [77]. Notably, the polo-kinase inhibitor GW843682X was found to alter flagellar length/biogenesis and potentially modulate cytokinesis [78].

Furthermore, mavelertinib, a tyrosine kinase inhibitor (EGFR-TKI), exhibited effects in a mouse infected with *G. intestinalis* [79]. Another compound tested was azidothymidine (AZT), an antiretroviral agent that is an efficient substrate for the parasite thymidine kinase, effectively inhibiting parasite DNA synthesis and proliferation [80]. These findings and other data demonstrate the distinctive characteristics of parasite kinases and their appeal as targets for anti-*Giardia* drug discovery.

### 4.7. Histone Acetylation as a Targeted Drug

*G. intestinalis* histones are susceptible to post-translational modifications, including acetylation, an important epigenetic mechanism observed in various eukaryotes. This process plays a regulatory role in the parasite’s genome, with the activity of histone acetyltransferase and histone deacetylase enzymes governing it. The levels of acetylation on histones are associated with the control of gene expression during the differentiation of trophozoites into cysts and the regulation of transcription and expression of genes encoding specific variant surface proteins (VSPs) [81,82,83].

Several histone deacetylase inhibitors are widely used in anticancer treatments and are available in the therapeutic field [84]. Through drug repositioning efforts, several studies have demonstrated the potential of these compounds against protozoan parasites, including *G. intestinalis*. For instance, Trichostatin A, tubastatine A, and nicotinamide have been shown to inhibit trophozoite proliferation [82,83,85]. Additionally, novel HDAC inhibitors named KH-TFMDI, a member of the 3-arylideneindolin-2-one family, and KV-46, a 4-[(10H-phenothiazin-10-yl)methyl]-N-hydroxy benzamide, exhibited the ability to modulate *G. intestinalis* growth, with IC50 values below 1 μM (Figure 17). Moreover, these compounds induced cytokinesis arrest and the formation of vacuoles with autophagic characteristics [67,86]. Although the primary mechanism of action for these compounds involves the inhibition of histone deacetylases, it is possible that they also impact other parasite pathways.

### 4.8. G. intestinalis Redox Metabolism

*G. intestinalis* is a microaerophilic protozoan that can tolerate the low concentrations of oxygen typically found in the small intestine. However, its metabolism relies on several enzymes sensitive to this molecule. For instance, enzymes in the glycolytic pathway, such as PFOR, play a crucial role in the parasite’s energy production, as it lacks conventional mitochondria. Molecular oxygen and reactive oxygen species can react with the FE-S cluster of these enzymes, necessitating the presence of mechanisms that eliminate toxic substances within the parasite. Notably, enzymes such as superoxide reductase, peroxiredoxins, thioredoxin reductase, NADH oxidase, and diaphorase have been identified and are involved in the redox metabolism of the parasite [87,88,89]. It is worth mentioning that the antioxidant glutathione has not yet been detected in *G. intestinalis*, with cysteine serving as the primary cellular redox buffer [89].

Given the distinct nature of the parasite’s enzymes compared to those found in higher eukaryotic cells, the redox metabolism pathway presents an enticing target for drug development, especially considering its crucial role in eliminating toxic elements. Within this context, thioredoxin reductase emerges as an intriguing target despite its functional role not being fully understood. Over the past few decades, studies have demonstrated that this enzyme plays a role in reducing metronidazole, the most used drug in cases of *Giardiasis*. Simultaneously, it is also a target for intermediate metabolites produced during the drug reduction process [32]. Additionally, auranofin, an antirheumatic compound, has been investigated through repositioning studies and has shown promising results against *G. intestinalis*, even reaching phase I of clinical trials [90]. Furthermore, the anti-tumoral agent NBDHEX, known for its inhibitory effects on human glutathione-S-transferases, also targets thioredoxin reductase and other molecules within the parasite [91].

**Table 2 pathogens-12-00810-t002:** Potential targets of drugs affecting *G. intestinalis* used in experimental chemotherapy studies.

Potential Target	Drug	In Vitro Effects on the Parasite	Reference
Cytoskeleton—Microtubules	Nocodazole Colchicine Taxol Oryzaline Mebendazole	Ventral disc disrupted Flagellar length altered Flagella bending Cytokinesis blocked Median body size altered	[58,59,60,61]
Cytoskeleton—Actin	Cytochalasin D Latrunculin A Jasplakinolide	Flagella internalization Multiflagellated cells Unshaped parasite Endocytosis	[62,63]
Organelles	Metronidazole Pyrantel pamoate Azasterol	Peripheral vesicles increased	[34,64]
Cell death	Beta-lapachone Linearolactone Kaempferol Lactoferrin peptides and 1 hydroxy-2-hydroxymethyl anthraquinone Metronidazole KH-TFMDI	Chromatin condensation Vacuolization Membrane blebbing	[66,67]
Carbohydrate metabolism	Disulfiram Omeprazole	Lamellar structures Vacuoles Increase in glycogen deposition	[70,71]
Lipid metabolism	24,25-(R,S)-epiminolanosterol Azasterol Bisphosphonates	Autophagic-like vacuoles Induce encystation	[73,74]
Kinases	Aurora kinase inhibitors BKI1213 GW843682X Polo-like kinases specific inhibitor Mavelertinib	proliferation compromitted cytokinesis flagellar length/biogenesis altered	[76,77,78,79,80]
Histone acetylation	Trichostatin A Tubastatine A Nicotinamide KH-TFMDI [(10H-phenothiazine-10-yl)methyl]-N-hydroxybenzamide	Cytokinesis arrest Autophagic-like vacuoles	[81,82,83,85,86]
Redox metabolism	Metronidazole Auranofin NBDHEX	DNA fragmentation Peripheral vesicles increased	[34,90,91]

## 5. New Compounds

Given the current issues with compounds used to treat *Giardiasis*, there is an ongoing effort to develop new chemotherapeutic alternatives. Several approaches are being explored, including: (a) attempts to utilize compounds that exhibit activity against other anaerobic protists, such as *Trichomonas vaginalis* and *Entamoeba histolytica*; (b) testing compounds that have inhibitory effects on enzymes involved in metabolic cycles observed in other cells are also present in *G. intestinalis*; (c) repositioning compounds already used for other diseases; and (d) extracts and molecules derived from plants, animal venoms, and, more recently, from marine organisms. These strategies increasingly benefit from high-performance virtual screening and high-throughput screening of new compounds in molecule libraries. The following briefly describes recent efforts to identify new compounds with activity against *Giardia*.

### 5.1. Repositioning Compounds—Drugs Used for Other Diseases

Robenidine is an anticoccidial drug first developed in the 1970s and now used regularly. In addition, it is added to the diet of various economically important animals, such as chickens and rabbits. Chemically, it is classified as a guanidine and has demonstrated inhibitory activity against the growth of *G. intestinalis*, with an IC50 in the range of 1 µm. Several analogs have been synthesized, and a library of 275 compounds has been developed and patented. Some of these derivatives, specifically compounds 45, 47, 48, and 49, have exhibited IC50 values below 10 µM. Additionally, certain derivatives have shown an ability to inhibit the adhesion of the trophozoite to the substrate. This action is particularly interesting as the pathogenicity of the protozoan largely relies on its capacity to adhere to the surface of intestinal epithelial cells [92,93].Secnidazole has been reported by Cheung et al. [94] as a treatment option for *Giardiasis*.Quinacrine has demonstrated promising efficacy in human cases of nitroimidazole-refractory Giardiasis [95,96].As mentioned in the previous section, azidothymidine (AZT), an antiretroviral drug, has activity against Giardia, including strains resistant to metronidazole. In addition, AZT has shown inhibitory effects on cyst formation and, in experimental trials conducted on infected gerbils, reduced the number of trophozoites and cysts [80].Disulfiram has been extensively tested against various pathogens and has also shown activity against *Giardia* [70,97].As mentioned in the previous section, Auranofin, an antirheumatic compound, showed promising results against G. intestinalis [90].Omeprazole and a series of 2-mercapto benzimidazoles, which are derivatives of omeprazole, have effects on *Giardia*. Several compounds within this series displayed IC50 values in the range of 17 µM while exhibiting low toxicity toward mammalian cells [71,98].Mavelertinib exhibited substantial growth inhibition at a concentration of 5 µM. Furthermore, it demonstrated a significant effect in experimental Giardiasis, resulting in a notable reduction in parasite presence. This effect was evaluated through non-invasive imaging of whole mice infected with a *Giardia* green beetle strain [79].Steroid hormone 20-hydroxyecdysone showed efficacy against human *Giardia* infection. Furthermore, this drug exhibited superior effectiveness to metronidazole, as 4% of patients resisted the classical compound [99].Fumagillin, an anti-microsporidiosis drug, showed effects in a giardiasis mouse model. [100].Orlistat, a pancreatic lipase inhibitor used for treating obesity, demonstrated in vitro activity against *G. intestinalis* [101].

### 5.2. Repositioning Compounds—Compound Screening Library and Other Synthetic Drugs

A library of 130 quinoxaline 1,4-di-N-oxides and certain derivatives within this library has demonstrated growth inhibitory activity with IC50 values below 10 µM. Notably, compounds 50, 51, 52, 53, and 54 were subjected to experimental infections in mice, and they exhibited superior activities compared to metronidazole and nitazoxanide [102,103].Triazoxins, a class of novel nucleoside analogs, have exhibited noteworthy activity against *Giardia*. Certain compounds within this group have demonstrated the ability to inhibit trophozoite growth with an IC50 of 5 µM. Additionally, other analogs have shown the capacity to inhibit trophozoite–cyst transformation in vitro [104].A screening of a substantial number (2451) of compounds from the Australian Scaffolds Library identified 40 hits against *Giardia*, exhibiting an IC50 of approximately 10 µM after 48 h of cultivation. Among these hits, three compounds demonstrated an IC50 of 1 µM, while CL9406 exhibited the lowest IC50 at 180 nM. Notably, the compound SN00797640 displayed potent activity against assemblages A, B, and metronidazole-resistant parasites [105]. However, further analysis of these compounds is required.Recently, Zheng et al. [106] reported the remarkable potency of a 3-nitroimidazo[1,2-b]pyridazine compound with an IC50 in the nanomolar (nM) range. This finding highlights the significance of this particular compound and warrants special attention.Due to the high glycolytic activity of *Giardia*, researchers have explored the testing of triose phosphate isomerase inhibitors. Compounds including benzothiazole, benzoxazole, benzimidazole, and sulphydryl derivatives have demonstrated meaningful activity in this regard [107].Interestingly, an inhibitor targeting the fused proteins glucose-6-phosphate dehydrogenase and 6-phosphogluconolactonase was observed with an IC50 of 10 µM [108].Benzopyrrolizidines have been successfully tested in *Giardia* cultures. Seventy-four compounds were evaluated, with several demonstrating an IC50 of 11 µM. These compounds induced notable morphological alterations in the parasite, including the loss of nuclei. Therefore, further detailed analysis of these c-mercapto benzimidazole compounds is warranted [109].Fernandez-Lainez et al. [110] reported the efficacy of compounds targeting arginine deiminase, an enzyme involved in a metabolic pathway associated with ATP synthesis, against *Giardia*.Gold nanoparticles, known for their broad-spectrum microbicidal activity, have demonstrated effectiveness against experimental *Giardiasis* in rats. These nanoparticles effectively inhibit the proliferation of trophozoites in the small intestine and the formation of cysts. The effects were evaluated using light and electron microscopy, revealing intestinal cell lesions’ recovery [111].The association of chitosan nanoparticles with metronidazole also exhibited significant efficacy in treating experimental *Giardiasis* in hamsters [112].In a recent study, Zoghroban et al. [113] demonstrated the significant efficacy of L-citrulline in controlling experimental *Giardiasis* in rats. The treatment resulted in reduced trophozoites in the intestinal mucosa and the complete elimination of cysts in the stool.

### 5.3. Natural Compounds

On the other hand, natural compounds have also been tested, primarily targeting *T. vaginalis*. One comprehensive analysis focused on a polyphenolic extract derived from *Punica granatum* peel, which induced cytoskeletal changes in *Giardia* trophozoites. Within the extract, ellagitannins, flavones, and ellagic acid exhibited inhibitory effects on cell growth and adhesion [114]. Moreover, the extract altered the expression and distribution of alpha-tubulin in trophozoites, as confirmed via Western blot and immunofluorescence microscopy. Scanning electron microscopy revealed altered cell morphology, shifting from the typical pear shape to a more elongated form, along with protrusions on the dorsal surface.

A study documented the examination of hydrolases secreted by the probiotic bacterium *Lactobacillus johnsonii* La1, revealing its toxicity towards *G. intestinalis* [115]. Additionally, research has demonstrated that the probiotics *Saccharomyces boulardii* and *S. bouvarvardias* significantly decreased the prevalence of *Giardiasis* infection in gerbils and humans, respectively [115,116].

In a controlled experiment, Daflon, a natural product used as a nutrient supplement and an antidiabetic drug, was evaluated alone or in combination with metronidazole in mice infected with Giardia. The findings demonstrated the efficacy of Daflon [117]. Drinic et al. [118] reported the utilization of 14 compounds isolated from seven classes. Among these compounds, methylgerambullin exhibited anti-*Giardia* activity at a concentration of 14.6 µM. Furthermore, synthesizing this compound enables the potential development of novel derivatives.

## 6. Conclusions

The data presented in this review highlight a series of promising new compounds that warrant special attention. Most of the published results are derived from in vitro experiments, where microscopy techniques are an important approach in the examining of morphological changes, identifying cell death processes, and predicting the mechanisms of action of these molecules. It is crucial for more frequent in vivo experiments to be conducted, preferably utilizing the gerbil as an experimental model [119], which is particularly important for compounds that exhibit an in vitro effect with IC50 values below 10 µM and demonstrate low toxicity towards human cells in culture. Furthermore, advancing knowledge regarding basic biological processes in Giardia necessitates adopting new methodology-based approaches. These approaches include the identification of metabolite transporters for various organelles, particularly peripheral vacuoles and mitosomes, through utilizing genome-wide CRISPR-based forward genetic screens, untargeted and targeted metabolomics, HDX-MS, and DNA-barcoded chemical screens.

Lastly, it remains imperative for funding agencies dedicated to scientific research in the field of health to initiate specific programs aimed at identifying new drug targets and developing novel drugs against pathogenic organisms responsible for neglected diseases.

## Figures and Tables

**Figure 1 pathogens-12-00810-f001:**
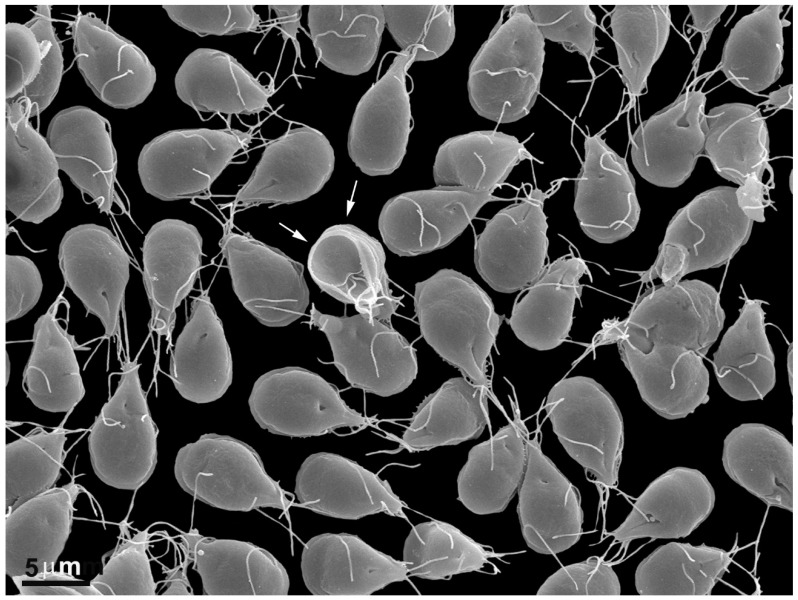
Scanning electron microscopy of *Giardia intestinalis* trophozoites adhered to the substrate. Notice that almost all parasites are seen by their dorsal view. The arrows point to a cell in the ventral view where the disc can be seen. Benchimol, unpublished.

**Figure 2 pathogens-12-00810-f002:**
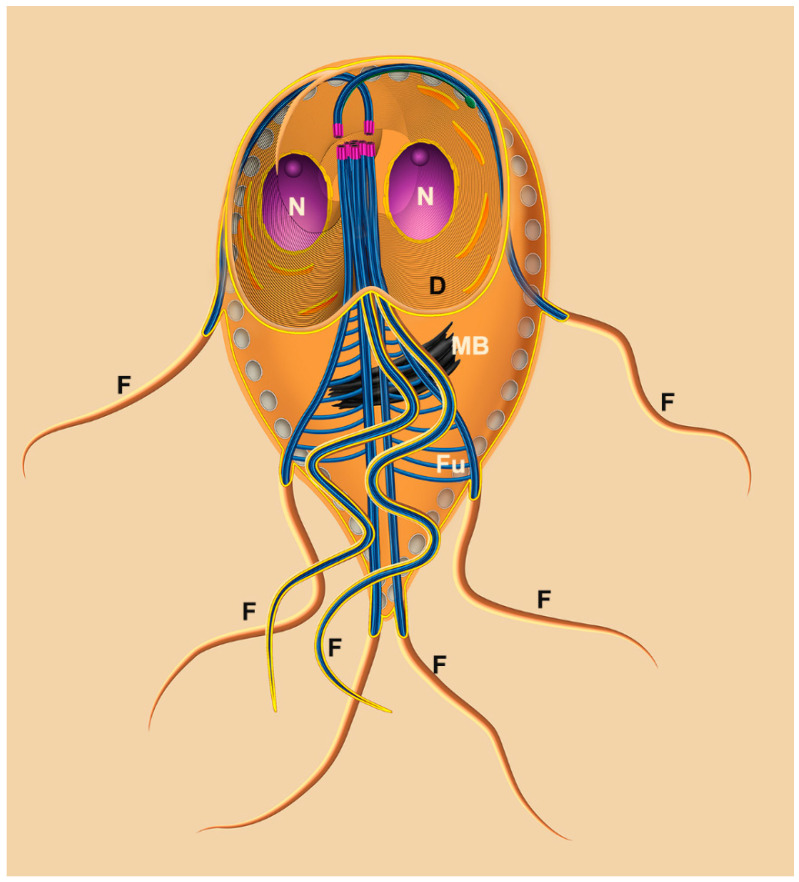
Scheme of *G. intestinalis.* The trophozoite displays two nuclei (N), peripheral vesicles (PVs), a ventral disc (D), median body (MB), funis (Fu), and four flagella pairs. Reproduced from [6].

**Figure 3 pathogens-12-00810-f003:**
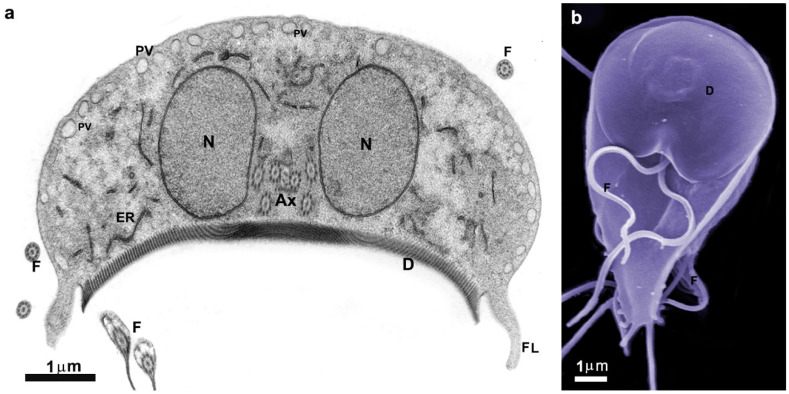
Micrograph of *Giardia intestinalis* trophozoites as observed using transmission (**a**) and scanning electron microscopy (**b**). Nuclei (N), ventral disc (D), peripheral vesicles (PVs), axonemes (Axs), flagella (F), profiles of the endoplasmic reticulum (ER), and flange (FL). In (**b**), the parasite is on a ventral side, where the disc (D) and the flagella can be observed. Benchimol, unpublished.

**Figure 4 pathogens-12-00810-f004:**
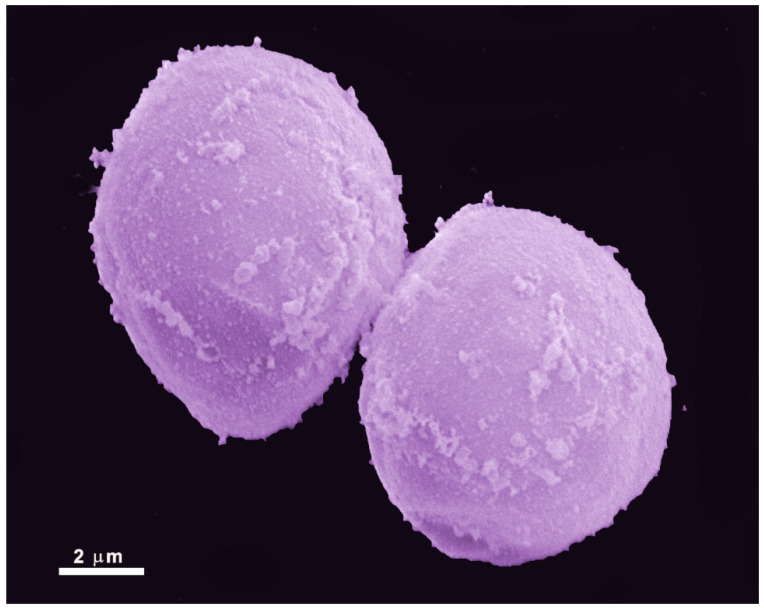
Scanning electron microscopy of *G. intestinalis* cysts. Benchimol, unpublished.

**Figure 5 pathogens-12-00810-f005:**
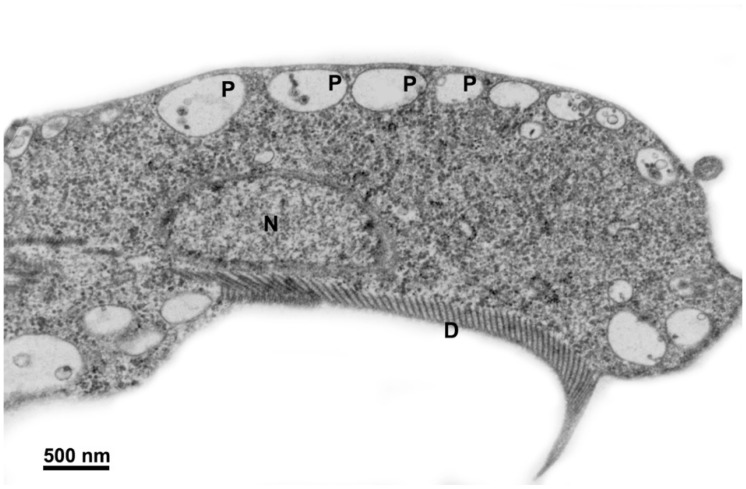
Micrograph of *Giardia intestinalis* trophozoites treated with 5 µg/mL of metronidazole for 24 h and observed via transmission electron microscopy. It is possible to note the enlargement of peripheral vesicles (P). N, nucleus; D, ventral disc. Gadelha, unpublished.

**Figure 6 pathogens-12-00810-f006:**
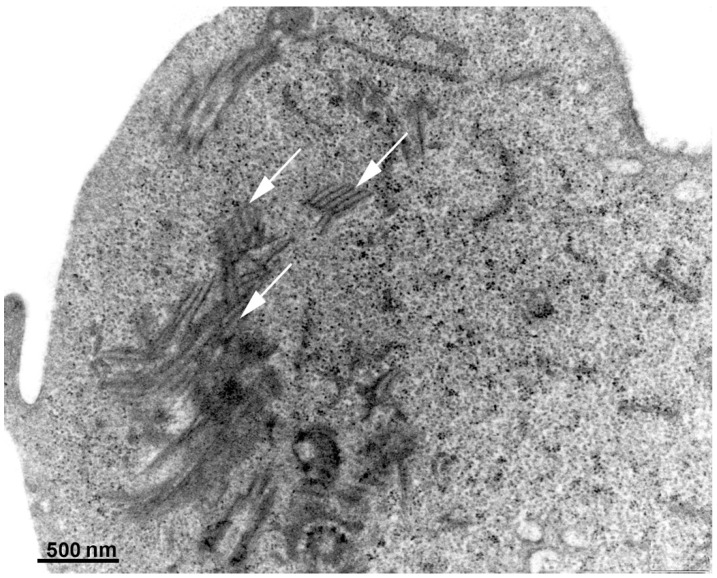
*Giardia intestinalis* trophozoites as observed using transmission electron microscopy after treatment with 1 µg/mL of albendazole for 24 h. Note that the ventral disc is fragmented (arrows). Gadelha, unpublished.

**Figure 7 pathogens-12-00810-f007:**
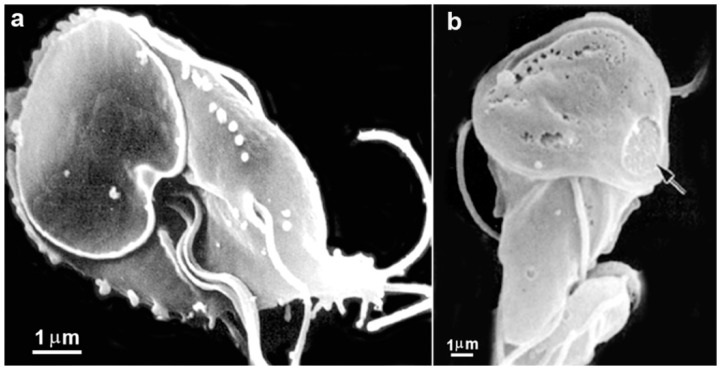
Scanning electron microscopy of *G. intestinalis* after treatment with nitazoxanide at 1 µg/mL (**a**) and 3 µg/mL (**b**) for 24 h. It is possible to note plasma membrane blebbing (**a**) and damage in the ventral disc (arrow) (**b**). Reproduced from [48].

**Figure 8 pathogens-12-00810-f008:**
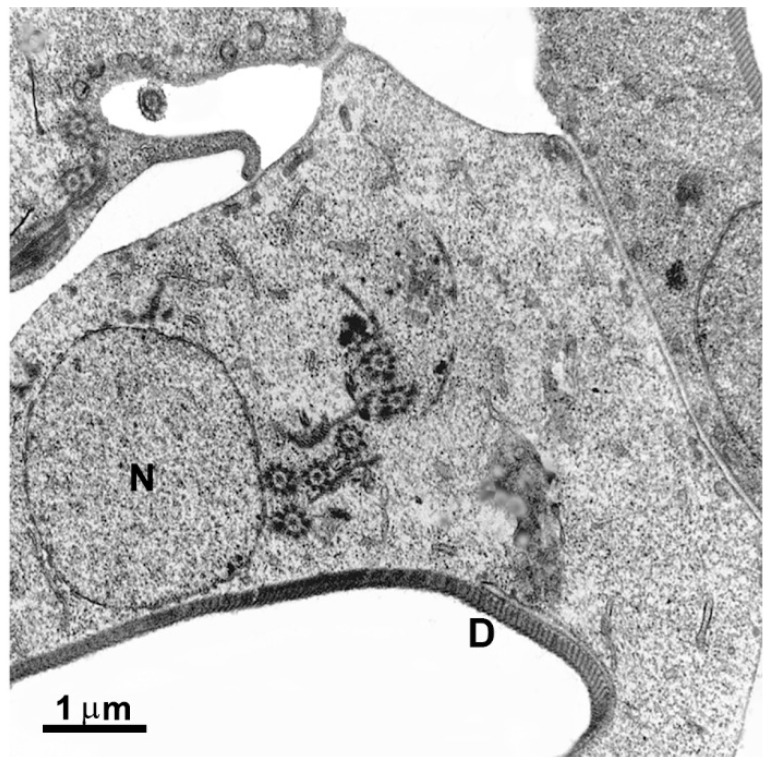
*G. intestinalis* trophozoites were treated with 1 μg/mL of furazolidone for 24 h. The cytoplasm presents a depletion of glycogen granules and ribosomes. D, ventral disc. N, nucleus. Gadelha, unpublished.

**Figure 9 pathogens-12-00810-f009:**
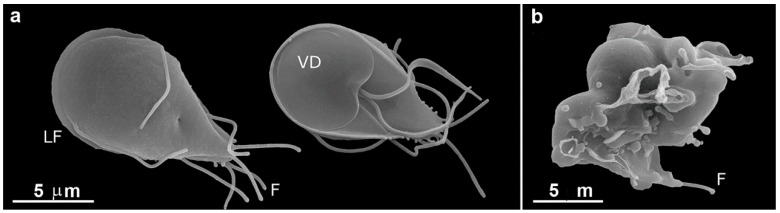
Scanning electron microscopy of *G. intestinalis* after 10 µM nocodazole treatment. (**a**) A typical untreated cell displays its dorsal and ventral views. Notice its pear shape, four pairs of flagella (F), lateral flange (LF), and adhesive disc in the ventral face (VD). (**b**) After 10 µM nocodazole treatment for 24 h, the cells became completely misshapen, presenting abnormal flagella number, irregular dorsal surface, membrane blebs, and loss of normal morphology. Reproduced from [58].

**Figure 10 pathogens-12-00810-f010:**
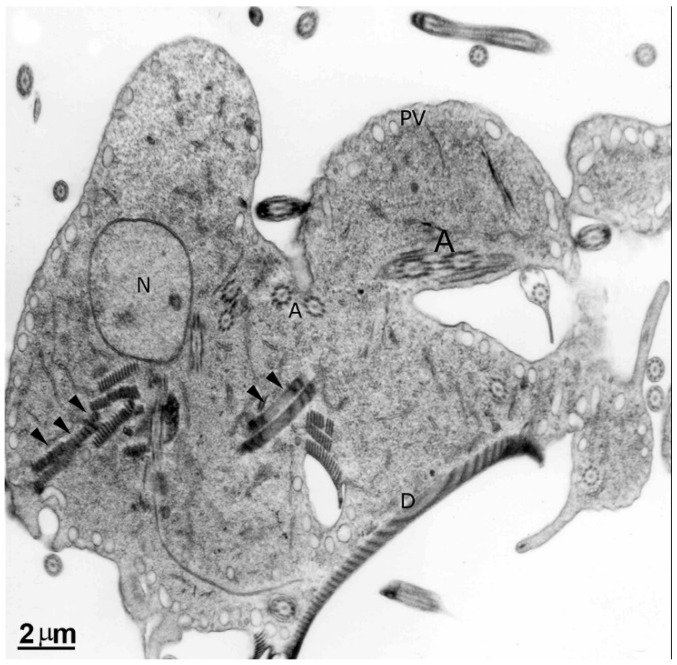
Transmission electron microscopy of *G. intestinalis* after 200 µM colchicine treatment for 24 h. Note that cells are misshapen, present disc fragmentation (arrowheads), and cytokinesis arrested. PVs, peripheral vesicles, N, nucleus, D, ventral disc, A, flagella axoneme. Reproduced from [58].

**Figure 11 pathogens-12-00810-f011:**
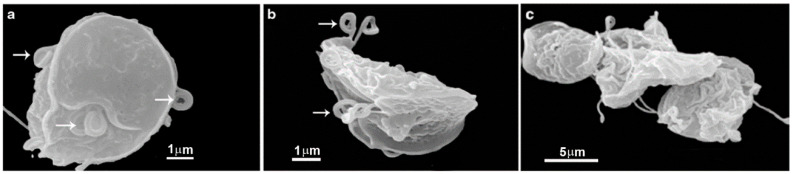
*G. intestinalis* treated with 50 µM oryzalin for 24 h. The parasites present shortening and curling of the flagella (arrows in (**a**,**b**)) and blockage of the cell division process (**c**). Reproduced from [61].

**Figure 12 pathogens-12-00810-f012:**
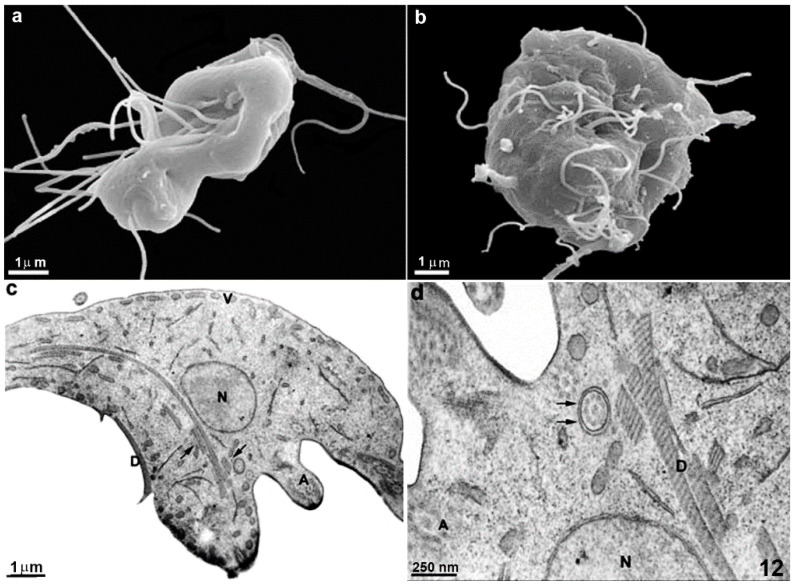
Scanning (**a**,**b**) and transmission electron microscopy (**c**,**d**) of *G. intestinalis* treated with 10 μM (**a**) and 1 μM cytochalasin D (**b**–**d**) for 24h. In (**a**,**b**), it is possible to note that the cells present flagella displacement and bizarre shapes. In (**c**,**d**), the ventral disc (D) is fragmented, and some flagella are internalized (arrows). N, nucleus, V, peripheral vesicles. Reproduced from [63].

**Figure 13 pathogens-12-00810-f013:**
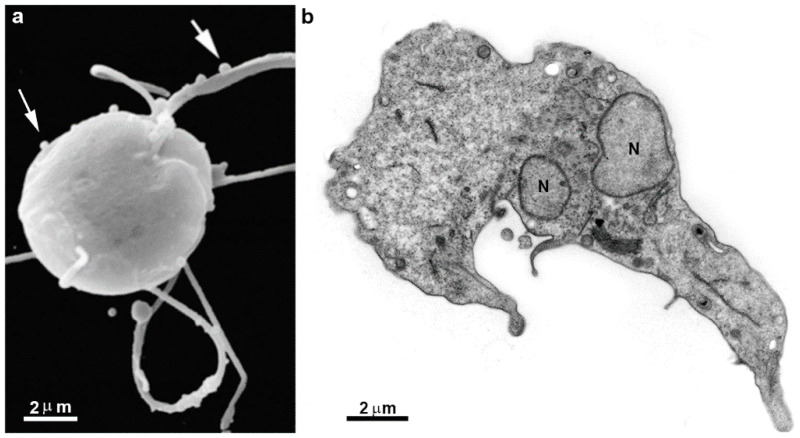
Scanning (**a**) and transmission electron microscopy (**b**) of *G. intestinalis* after treatment with 10 μM β-lapachone for 3 h. The parasite is mishappened, with plasma membrane and flagella surface blebs (arrows) and several organelle alterations (**b**). N, nucleus. Image (**a**), after [66]; Image (**b**), Benchimol, unpublished.

**Figure 14 pathogens-12-00810-f014:**
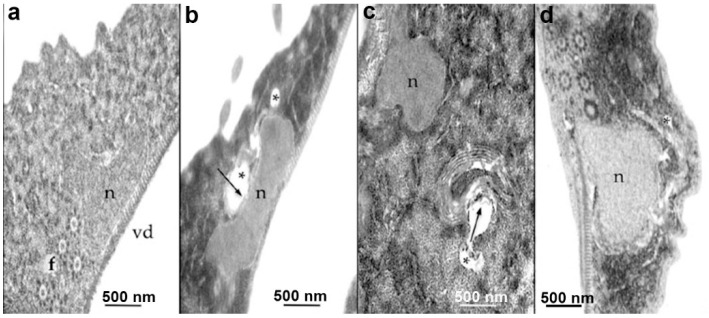
*G. intestinalis* control (**a**) and treated with omeprazole 0.072 µM (**b**), 0,144 µM (**c**), and 1 µM (**d**). After treatment, vesicles (asterisks) and lamellar structures (arrows) are observed. N, nucleus; VD, ventral disc. Reproduced from [71].

**Figure 15 pathogens-12-00810-f015:**
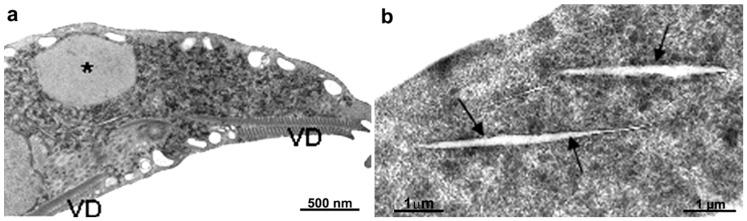
Transmission electron microscopy images of *G. intestinalis* treated with 1 mM azasterol for 24h. (**a**) Notice the enlargement of the peripheral vesicles (P) and encystation-specific vesicles (*). (**b**) Clefts (arrows) in the endoplasmic reticulum are also observed after treatment. VD: ventral disc. Reproduced from [73].

**Figure 16 pathogens-12-00810-f016:**
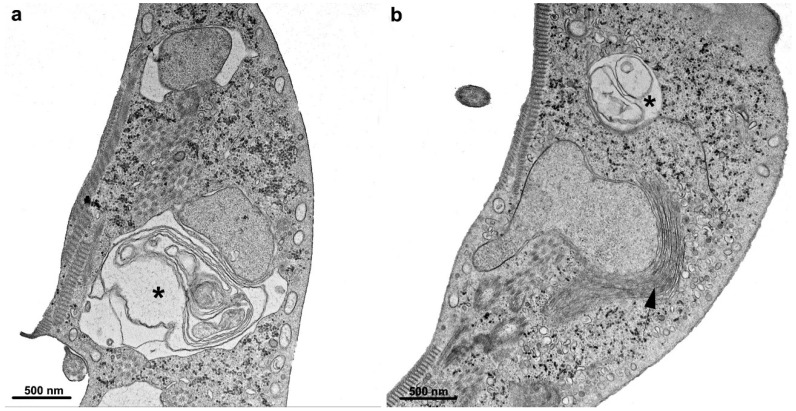
Transmission electron microscopy of *G. intestinalis* treated with 300 μM ibandronate for 48 h. (**a**,**b**) Vacuoles with lamellas (*) are observed in the cytoplasm. Note that myelinic figures (arrow) are close to the nucleus. Reproduced from [74].

**Figure 17 pathogens-12-00810-f017:**
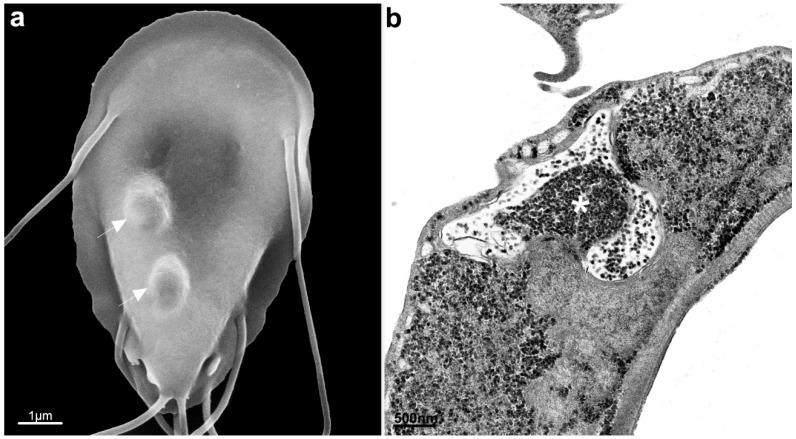
*G. intestinalis* treated with 5 μM histone deacetylase inhibitor KV-46 for 48 h. (**a**) Scanning electron microscopy shows protrusions on the treated parasite’s dorsal surface (arrows). (**b**) Transmission electron microscopy: vacuoles containing lamellar figures and cytoplasmic granules (*) are observed on cells treated with KV-46. Gadelha, unpublished.

## Data Availability

Not applicable.

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
