# Peer review of "Ultrastructural Alterations of the Human Pathogen Giardia intestinalis after Drug Treatment"

_pathogens, 2023, doi:10.3390/pathogens12060810_

Round 1

Reviewer 1 Report

Comments to the manuscript entitled, “Ultrastructural alterations of the human pathogen Giardia intestinalis after drugs treatment” by Marlene Benchimol, Ana Paula Gadelha and Wanderley de Souza

In general, the manuscript include an extensive review of the works related to the effect and ultrastructural alterations on mainly Giardia intestinalis trophozoites by various drugs. However, there is a need to consider all the comments on the manuscript thus a better description of the works that the authors include in the manuscript could be achieved.  

Comments:

Lane 13. Most of the studies included in the manuscript are from studies in vitro. Thus the term in vivo should be deleted.  

Lanes 22-24. The sentences are not well connected. This part of the manuscript has to be re-written.

Lanes 25-27. The last paragraph of the abstract does not seem to include conclusions. Rather it includes general ideas. Thus it needs to be corrected.

Lane 60. The figure legend should be corrected as follows: Micrograph of Giardia intestinalis trophozoites adhered to the substrate as observed by scanning electron microscopy.

Lane 64. The sentence should be change to: Morphological description of the two sages in the life cycle of Giardia intestinalis as observed by light and electron microscopy.  

Lanes 65-66. As the sentence is written it seems that there are three forms and not two stages in the life cycle of the parasite. This should be change to “two developmental stages: the cyst and the trophozoite”

Lane 83. Figure 3. The figure legend should be changed as follows: Micrograph of Giardia duodenalis trophozoites as observed by transmission (a) and scanning electron microscopy.

Lane 72. The sentence should be change to “The trophozoite measures 9 to 21 microns (μm) long and 5 to 15 μm wide and is 2 to 4 μm thick.”

Lane 135. The scale bar is missing on the micrograph.

Lane 166. The sentence seems to indicate that the peripheral vesicles increase in number, but according to the micrographs the vesicles increase on size. Thus the sentence should be corrected.

Lane 176. The sentence should be change to: Micrograph of Giardia intestinalis trophozoites treated with 5 µg/ml of metronidazole for 24 hours and observed by transmission electron microscopy.

 Lanes 197-198. A mechanism involved in the conversion of Albendazole into sulfoxide and Sulfone has been recently described (Pech-Santiago et al., PLOs Pathogens 2022). Thus a reference to this work needs to be included.

Lane 206.  The figure legend should be change as follows: Giardia intestinalis trophozoites as observed by transmission electron microscopy after treatment with 1 mµ/ml albendazole for 24 hrs.

Lane 219. Include references to the work referred.

Lane 224. Adhesion of trophozoites may be affected however no evidences are included in this paragraph for an effect on absorption. This needs to be corrected.

Lane 230. There is a need to include the concentration of nitoxozanide when a higher concentration is referred.

Lane 246. The ventral disk should be marked in the micrograph.

Lane 295. M should be corrected to µM

Lane 336. Add (c-d) after transmission electron microscopy.

Lane 319. After the word analysis, change and add the following:  thus these drugs are more suitable for the treatment of giardiasis.

Lane 331. In the sentence after 10 µM cytochalasin change the following: which binds to actin inhibiting the polymerization and elongation of microfilaments to a greater extend.

Lanes 341-361.  In section 4.4 Drugs inducing cell death: Beta-lapachone and curcumin The text should be change and the authors need to explain which drugs induce cell death. Several published studies have shown that apoptotic-like cell death in Giardia trophozoites is induced by a plethora of compounds in a way preceded by processes not only related to membrane blebbing but others that include: DNA fragmentation, ROS production or phosphatidylserine exposure at outer cell membrane. These studies have been reported, thus the authors need to review these studies and include in the manuscript the reported findings.  Indeed, the inclusion of only two compounds provides a limited landscape in the context of the review presented.

Lane 384. Change the font of the words “Cholesterol is”  

Lane 397. Please clarify if you refer to "Encystation Specific Vesicles”

Lane 443. There is no reference to the figure in the text.

Lane 451. Delete the word gas

Lane 483. Change: aiming the determination of IC50 for “aiming to determine of the IC50”

Lane 484. Change detached for death cell

Lane 593. It would be important that the authors support why they propose to use gerbils as experimental model.  

Lanes 513-516.  The description of the text is confusing because Paromomycin is a recognized second-line agent for treatment of giardiasis, particularly in pregnant women. The need for "more adequate studies" is not clearly supported in the text. Thus the authors need to re-write this part of the text.

Lanes 492-589.  The list of novel compounds under testing against Giardia is quite long and includes compounds from several sources (natural, semi-synthetic and synthetic). This section of the manuscript should be connected to the previous paragraph and re-organized to provide a more structured and comprehensive view of the drugs that could be used against Giardia and the current strategies to do so.

The numbers included in the micrographs are confusing, therefore these should be deleted.

Finally, it is important that the authors emphasize the importance and impact  of determining the effects of various drugs on Giardia and their analysis by transmission and scanning electron microscopy. 

The manuscript needs to be reviewed for English grammar and spelling.

Author Response

Answers to Referee 1:

Lane 13. Most of the studies included in the manuscript are from studies in vitro. Thus the term in vivo should be deleted.  

Answer: Thanks. It was deleted.

Lanes 22-24. The sentences are not well connected. This part of the manuscript has to be re-written.

Answer: Thanks. It was modified.

Lanes 25-27. The last paragraph of the abstract does not seem to include conclusions. Rather it includes general ideas. Thus it needs to be corrected.

Answer: Thanks. It was modified.

Lane 60. The figure legend should be corrected as follows: Micrograph of Giardia intestinalis trophozoites adhered to the substrate as observed by scanning electron microscopy.

Answer: Thanks. It was changed.

Lane 64. The sentence should be change to: Morphological description of the two sages in the life cycle of Giardia intestinalis as observed by light and electron microscopy.  

Answer: Thanks. It was changed.

Lanes 65-66. As the sentence is written it seems that there are three forms and not two stages in the life cycle of the parasite. This should be change to “two developmental stages: the cyst and the trophozoite”

Answer: Thanks. It was changed.

Lane 83. Figure 3. The figure legend should be changed as follows: Micrograph of Giardia duodenalis trophozoites as observed by transmission (a) and scanning electron microscopy.

Answer: Thanks. It was changed.

Lane 72. The sentence should be change to “The trophozoite measures 9 to 21 microns (μm) long and 5 to 15 μm wide and is 2 to 4 μm thick.”

Answer: Thanks. It was changed.

Lane 135. The scale bar is missing on the micrograph.

Answer: Thanks. It was changed.

Lane 166. The sentence seems to indicate that the peripheral vesicles increase in number, but according to the micrographs the vesicles increase on size. Thus the sentence should be corrected.

Answer: Thanks. It was changed.

Lane 176. The sentence should be change to: Micrograph of Giardia intestinalis trophozoites treated with 5 µg/ml of metronidazole for 24 hours and observed by transmission electron microscopy.

Answer: Thanks. It was changed.

 Lanes 197-198. A mechanism involved in the conversion of Albendazole into sulfoxide and Sulfone has been recently described (Pech-Santiago et al., PLOs Pathogens 2022). Thus a reference to this work needs to be included. ANA

Answer: Thanks. The reference was included as suggested.

Lane 206.  The figure legend should be change as follows: Giardia intestinalis trophozoites as observed by transmission electron microscopy after treatment with 1 mµ/ml albendazole for 24 hrs.

Answer: Thanks. It was changed.

Lane 219. Include references to the work referred.

Answer: The reference was not included because the text was modified.

Lane 224. Adhesion of trophozoites may be affected however no evidences are included in this paragraph for an effect on absorption. This needs to be corrected  

Answer: Thanks. The text was re-written.

Lane 230. There is a need to include the concentration of nitoxozanide when a higher concentration is referred.

Answer: The concentration was included in the legend as suggested.

Lane 246. The ventral disk should be marked in the micrograph.

Answer: Thanks. It was marked.

Lane 295. M should be corrected to µM

Answer: Thanks. It was corrected.

Lane 336. Add (c-d) after transmission electron microscopy.

Answer: Thanks. It was corrected.

Lane 319. After the word analysis, change and add the following:  thus, these drugs are more suitable for the treatment of giardiasis.

Answer: Thanks. It was changed.

Lane 331. In the sentence after 10 µM cytochalasin change the following: which binds to actin inhibiting the polymerization and elongation of microfilaments to a greater extend.

Answer: Thanks. It was changed.

Lanes 341-361.  In section 4.4 Drugs inducing cell death: Beta-lapachone and curcumin The text should be change and the authors need to explain which drugs induce cell death. Several published studies have shown that apoptotic-like cell death in Giardia trophozoites is induced by a plethora of compounds in a way preceded by processes not only related to membrane blebbing but others that include: DNA fragmentation, ROS production or phosphatidylserine exposure at outer cell membrane. These studies have been reported, thus the authors need to review these studies and include in the manuscript the reported findings.  Indeed, the inclusion of only two compounds provides a limited landscape in the context of the review presented.

Answer: The text was re-written and previous studies were included as recommended.

Lane 384. Change the font of the words “Cholesterol is”  

Answer: Thanks. It was corrected.

Lane 397. Please clarify if you refer to "Encystation Specific Vesicles”

Answer: Thanks. It was corrected.

Lane 443. There is no reference to the figure in the text.

Answer: Thanks. It was added.

Lane 451. Delete the word gas

Answer: Thanks. It was corrected.

Lane 483. Change: aiming the determination of IC50 for “aiming to determine of the IC50”

Answer: Thanks. It was changed.

Lane 484. Change detached for death cell

Answer: Thanks. It was changed.

Reviewer 2 Report

Comments for authors

Major points:

Table 1 and 2 are the same as previously published by Benchimol, Gadelha & de Souza 2022. There are only added three new drugs (Paramomycin, Quinacrine in the therapeutic drugs, and Azasterols, in the experimental chemoterapy) and others are missing. Moreover, the references are the same as the previous work although a few of them are even missing. I consider the authors need to give and extra-information of each drug to be worth adding those tables.

In addition, figures are also similar to those recently published by the same authors in their previous work.

Throughout the entire text, tables and foot of figures, the drugs appear with a lower or upper case letter indistinctly. Better uniform it.

The time of treatment with the drug appears only in some of the foot of figures. Better uniform it.

I consider that the present draft can be shortened and turned into a note instead of a full review.

Minor points:

Tittle: Giardia intestinalis should be written on italic.

Lines 14, 15 and 18: Giardia intestinalis should be written on italic

Line 25: writing error: parasite, e

Line 51: cyst can be present in contaminated water and food, but also in dirty hands….Add the main ways of infection.

Line 65-66: this text should be deleted. This sentence has the same idea as lines 49-50.

Line 74: complete the text: exit “in pairs” in different positions.

Line 86: Figure 3 text: add (F) after flagella

Line 90: complete the text: each presenting five chromosomes and similar amount of DNA

Line 95: Ventral disc: Are B-giardins and B-tubulins proteins involved in the ventral disc structure? Add explanation appropriately.

Line 102: Median body: Is there one or two median bodies? Is the shape of median body important? Add explanation appropriately.

Line 107: better scroll to the end: 150-200 nm approx.

Line 120-122: Is the whole sentence important? You are saying things that does not present. Can be removed.

Line 124: add an space in the parentheses before IscU

Line 124: After the reference [21] IscS and IscU they are repeated. Delete it.

Line 123 and 125: cluster assembly and complex biosynthesis of iron-sulfur (Fe-S) is also repeated. Correct the text.

Line 133: Cyst: better than “three” or four nuclei, mots common “two” or four nuclei. Correct the text.

Line 149 and 179: should not you put the formula?

Line 153: 5-nitroimidazole, called Metronidazole,

Line 158: medicine “against” giardiasis

Line 182-183: Is the whole sentence important? This sentence has the same idea as previous sentence lines 181-182.

Line 186: Mebendazole appears latter in Other compounds section (3.3). It is better to remove it here and change the text in line 185 to: Benzimidazoles carbamate “compounds” have the best activity….

Line 210: After the first sentence about Metronidazole and Albendazole, it could be useful to move forwards the text form the lines 317-319.

Line 210: The whole sentence “However, more recently, nitazoxanide has been suggested as an alternative to these drugs (Fig.7)” should be moved backwards to line 214. In addition, it would be better not to point to the Fig.7 in this sentence.

Line 214-215: the whole sentence can be deleted “Below, we will discuss some of them”.

Line 270-271: the whole sentence can be deleted “Below, we will……

Line 283: better delete “used clinically and described in more detail in the previous section”

Line 307: better put a new paragraph with Taxol, vinblastine….

Line 340: delete (TABLE 2) Section 4.3 refers to Table 2 but the text is not included in such a Table. The data in Table 2 are from section 4.4

Lines 384-387: very confusing sentence. Redraft.

Lines 389-391: very confusing sentence. Redraft.

Line 397: add (P) after peripheral vesicles. It will be also necessary to locate them in the TEM image.

Line 398: the reference 74 does not match with the one indicated in the text 73.

Line 485: a “basic step” is missing. We can read (a), (b), (c), (d), (f) and (g) but not (e).

Line 493: add colon:

Line 521: Giardia “com” IC50. What does it mean? With?

Line 525-526: rewrite the sentence: Recently, Zheng et al (102) reported a potent activity of a 3-nitroimidazole (1,2-b) pyridazine, in the nM range.

Line 553 and 558 and 588 and 595: is uM meaning the same as µM (line 549)? Standardize.

Line 597: Giardia on italic.

Author Response

Answers to Referee 2:

Comments for authors

 Major points:

TABELA Table 1 and 2 are the same as previously published by Benchimol, Gadelha & de Souza 2022. There are only added three new drugs (Paramomycin, Quinacrine in the therapeutic drugs, and Azasterols, in the experimental chemoterapy) and others are missing. Moreover, the references are the same as the previous work although a few of them are even missing. I consider the authors need to give and extra-information of each drug to be worth adding those tables.

Answer: We added new data. The tables were changed.

In addition, figures are also similar to those recently published by the same authors in their previous work.

Answer: The authors do not agree with the referee. Almost all figures are unpublished, and those published previously differ from the previous review.

Throughout the entire text, tables and foot of figures, the drugs appear with a lower or upper case letter indistinctly. Better uniform it.

Answer: We have done.

The time of treatment with the drug appears only in some of the foot of figures. Better uniform it.

Answer: Now, all figures present the time and drug concentration.

I consider that the present draft can be shortened and turned into a note instead of a full review.

Answer: The authors do not agree with the referee. This is an invited review, and when the invitation was done, we sent an abstract where the major points would be written, and the editor agreed.

Minor points:

Tittle: Giardia intestinalis should be written on italic.

Answer: When we submitted it was written in italics. Something is wrong with the copy.

Lines 14, 15 and 18: Giardia intestinalis should be written on italic

Answer: When we submitted it was written in italics. Something is wrong with the copy.

Line 25: writing error: parasite, e

Answer; Thanks, it was corrected.

Line 51: cyst can be present in contaminated water and food, but also in dirty hands….Add the main ways of infection.

Answer; Thanks, it was added.

Line 65-66: this text should be deleted. This sentence has the same idea as lines 49-50.

Answer; Thanks, it was removed.

Line 74: complete the text: exit “in pairs” in different positions.

Answer; Thanks, it was added.

Line 86: Figure 3 text: add (F) after flagella

Answer;  We did not find the missing because F (flagella) is already there.

Line 90: complete the text: each presenting five chromosomes and similar amount of DNA

Answer; Thanks, it was added.

Line 95: Ventral disc: Are B-giardins and B-tubulins proteins involved in the ventral disc structure? Add explanation appropriately.

Answer; Thanks, it was added.

Line 102: Median body: Is there one or two median bodies? Is the shape of median body important? Add explanation appropriately.

Answer; Thanks, it was added.

Line 107: better scroll to the end: 150-200 nm approx.

Answer: We did not understand this suggestion.

Line 120-122: Is the whole sentence important? You are saying things that does not present. Can be removed.

Answer: OK, it was removed.

Line 124: add an space in the parentheses before IscU

Answer: Thanks, it was corrected.

Line 124: After the reference [21] IscS and IscU they are repeated. Delete it.

Answer: Thanks, it was removed.

Line 123 and 125: cluster assembly and complex biosynthesis of iron-sulfur (Fe-S) is also repeated. Correct the text.

Answer: Thanks, it was corrected.

Line 133: Cyst: better than “three” or four nuclei, mots common “two” or four nuclei. Correct the text.

Answer: Thanks, it was corrected.

Line 149 and 179: should not you put the formula?

Answer: Thanks. We added all formulas in a new Table 1.

Line 153: 5-nitroimidazole, called Metronidazole

Answer: Thanks, it was corrected.

Line 158: medicine “against” giardiasis

Answer: Thanks, it was corrected.

Line 182-183: Is the whole sentence important? This sentence has the same idea as previous sentence lines 181-182.

Answer: The sentence was deleted.

Line 186: Mebendazole appears latter in Other compounds section (3.3). It is better to remove it here and change the text in line 185 to: Benzimidazoles carbamate “compounds” have the best activity.

Answer: Mebendazole information was removed from the last paragraph of section 4.1.

Line 210: After the first sentence about Metronidazole and Albendazole, it could be useful to move forwards the text form the lines 317-319.

Answer: Ok, it was corrected.

Line 210: The whole sentence “However, more recently, nitazoxanide has been suggested as an alternative to these drugs (Fig.7)” should be moved backwards to line 214. In addition, it would be better not to point to the Fig.7 in this sentence.

Answer: Ok, this was corrected.

Line 214-215: the whole sentence can be deleted “Below, we will discuss some of them”.

Answer: Thanks, it was deleted.

Line 270-271: the whole sentence can be deleted “Below, we will……

Answer: Thanks, it was deleted.

Line 283: better delete “used clinically and described in more detail in the previous section”

Answer: Thanks, it was deleted.

 Line 307: better put a new paragraph with Taxol, vinblastine….

Answer: Thanks, it was done.

Line 340: delete (TABLE 2) Section 4.3 refers to Table 2 but the text is not included in such a Table. The data in Table 2 are from section 4.4

Answer: Thanks. Now, the tables have been changed. Thus, the position is correct.

Lines 384-387: very confusing sentence. Redraft.

Answer: Thanks, it was done.

Lines 389-391: very confusing sentence. Redraft.

Answer: Thanks, it was done.

Line 397: add (P) after peripheral vesicles. It will be also necessary to locate them in the TEM image.

Answer: Thanks, it was done.

Line 398: the reference 74 does not match with the one indicated in the text 73. Answer: Thanks, the reference number was corrected.

Line 485: a “basic step” is missing. We can read (a), (b), (c), (d), (f) and (g) but not (e).

Answer: Thanks, it was corrected.

Line 493: add colon:

Answer: Thanks, it was added.

Line 521: Giardia “com” IC50. What does it mean? With?

Answer: Thanks, it was corrected. Yes.

Line 525-526: rewrite the sentence: Recently, Zheng et al (102) reported a potent activity of a 3-nitroimidazole (1,2-b) pyridazine, in the nM range.   

Answer: Thanks, it is now more clear.

 Line 553 and 558 and 588 and 595: is uM meaning the same as µM (line 549)? Standardize.

Answer: Thanks, it was corrected. Yes.

Line 597: Giardia on italic.

Answer: Thanks, it was corrected.

Lane 593. It would be important that the authors support why they propose to use gerbils as experimental model.

Answer: As requested we mentioned one example of a paper where gerbils were used as an experimental model of Giardia intestinalis infection.

We included: Peckova, R., Sak, B., Kvetonova, D., Kvac, M., Koritakva, E. & Foitova, I. 2018. The course of experimental giardiasis in Mongolian gerbil. Parasitol. Res. 117: 2437-2443 

Lanes 513-516.  The description of the text is confusing because Paromomycin is a recognized second-line agent for treatment of giardiasis, particularly in pregnant women. The need for "more adequate studies" is not clearly supported in the text. Thus the authors need to re-write this part of the text.

Answer: We deleted all reference to Paromomycin since it was already mentioned in a previous section.

Lanes 492-589.  The list of novel compounds under testing against Giardia is quite long and includes compounds from several sources (natural, semi-synthetic and synthetic). This section of the manuscript should be connected to the previous paragraph and re-organized to provide a more structured and comprehensive view of the drugs that could be used against Giardia and the current strategies to do so.

Answer: We do not agree with the suggestion since this a section dealing with recently reported compounds without connection with other sections.

The numbers included in the micrographs are confusing, therefore these should be deleted.

Answer: Thanks. They were deleted.

Finally, it is important that the authors emphasize the importance and impact  of determining the effects of various drugs on Giardia and their analysis by transmission and scanning electron microscopy.

 Answer: It is very well known and described in many papers, including several mentioned here, that electron microscopy allows the identification of structures and organelles affected by a certain compound, thus indicating potential targets for the compound.

Comments on the Quality of English Language

The manuscript needs to be reviewed for English grammar and spelling.

Answer: Thanks. It was reviewed.

Round 2

Reviewer 1 Report

please see the attached comments

Although the manuscript has been modified according to the comments included in the first review, the English grammar and spelling must still be revised by a Scientist whose native language is English.

Author Response

COMMENTS TO THE REVISED MANUSCRIPT ENTITLED AS: "Ultrastructural alterations of the human pathogen Giardia intestinalis after drugs treatment"

The revised version of the manuscript still need to be improved in the following specific points:

  1. The authors need to include in the abstract a reference of their findings of the work described in the manuscript in relation to the ultrastructural alterations that hey observed in Giardia after treatment with different drugs.

Answer: OK. The references were included in the abstracts as suggested.

  1. The sentence regarding the mechanism involved in conversion of Albendazole into sulfoxide and sulfone has been now included (lines 273-274). However, in the same paragraph (lines 280-283) the role of an antioxidant response in Albendazole resistance in this parasite (Argüello-García et al., 2015) is not mentioned and the ref. 43 in not included in this part of the text.

Answer: Thanks. The references were included.

  1. The description of New compounds against Giardia (lines 641-727) is quite long and it is again recommended to organize this part of the manuscript on the basis of types of compounds reported (repurposed, synthetic, natural, etc.) to provide a well- structured and concise view of these current strategies, as a Review warrants.

Answer: Ok. This section was shortened and organized in (a) Repositioning compounds – drugs used for other disease, (b) Repositioning compounds –compound screening library and other syntethic drugs, (c) Natural compound

  1. In the conclusions section there is no reference to the findings on the ultrastructural alterations observed in the parasite after treatment with different drugs. Therefore, the authors need to include in this section their conclusions of the work reported in the manuscript in relation to the effects observed in the parasite by the drugs tested.

Answer: The conclusion was included as suggested.

  1. Although the manuscript has been modified according to the comments included in the first review, the English grammar and spelling must still be revised by a Scientist whose native language is English.

Answer: The English grammar and spelling was revised as recommended. American Manuscript Editors corrected the manuscript, and we have paid for this service. A certificate is presented below.

Reviewer 2 Report

The reviewers comments have been introduced and now the manuscript has improved its quality. Now I recommend its publication.
